# A Risk-Aware Approach to Digital Procurement Transformation

**Željko Dudić** [1,*]‍, **Vijoleta Vrhovac** [1]‍, **Srđan Vulanović** [1], **Dušanka Dakić** [1], **Irma Erdeji** [2] **and Veselin Perović** [1]‍

1 Department of Industrial Engineering and Engineering Management, Faculty of Technical Sciences, University of Novi Sad, Trg Dositeja Obradovića 6, 21000 Novi Sad, Serbia; violeta.vrhovac@uns.ac.rs (V.V.); srdjanv@uns.ac.rs (S.V.); dakic.dusanka@uns.ac.rs (D.D.); vesa@uns.ac.rs (V.P.)

2 Faculty of Business Economics, Educons Unversity, Vojvode Putnika 87, 21208 Sremska Kamenica, Serbia; irmaing@gmail.com

* Correspondence: zeljko.dudic@yahoo.com

**Abstract:** As the digital procurement transformation continues to evolve, it is crucial to adopt a risk-aware approach to ensure successful outcomes. This paper delves into the necessity of a risk-aware approach to digital procurement transformation, specifically focusing on the role of professional procurement management and the significance of supplier partnerships in the digital environment. The research aimed to examine the critical role of risk management in the digital transformation of procurement. A total of 318 respondents from Serbia participated in the study. The role of professional management in procurement must encompass risk management to guarantee success in digital procurement transformation. Furthermore, the study identified that adequate preparation, information, and training for individuals involved are crucial for a seamless transition to digital procurement. The digital transformation of procurement contributes to sustainability by enhancing efficiency, transparency, and collaboration across the supply chain, ultimately fostering environmentally conscious practices and responsible resource management. In summary, the research underscores the need for a comprehensive and risk-aware approach to digital procurement transformation, considering the pivotal roles of procurement professionals, supplier partnerships, and the integration of new technologies.

**Keywords:** risk management; procurement; digital transformation; supplier; digital tools; training

## 1. Introduction

In recent years, digital procurement transformation has garnered significant attention from organizations looking to leverage digital technologies for streamlining procurement processes, enhancing data visibility, and gaining greater control over their supply chains [1]. The COVID-19 pandemic has further accelerated the implementation of digital procurement transformation, with organizations being compelled to shift to remote work and online procurement processes [2,3]. However, digital procurement transformation comes with its challenges, necessitating organizations to understand the potential risks and pitfalls associated with this transformation [4].

One of the primary challenges of digital procurement transformation is effectively managing the associated risks [3–6]. The adoption of new procurement technologies, systems, and processes exposes organizations to various risks such as cyber-attacks, data breaches, supply chain disruptions, and vendor management issues [3]. Moreover, the speed and complexity of digital transformation can introduce additional risks that organizations may not be adequately prepared to handle. To effectively manage these risks, procurement professionals must possess the necessary skills and knowledge to identify, assess, and mitigate risks [7].

To address these challenges, organizations need to adopt a risk-aware approach to digital procurement transformation. A risk-aware approach involves identifying and

mitigating risks associated with the transformation while maximizing the benefits of digital procurement transformation, requiring effective risk management practices.

The procurement function plays a crucial role in organizational success, especially in the context of digital procurement transformation [3]. The adoption of digital tools has transformed procurement processes, allowing organizations to streamline operations, enhance data visibility, and exercise greater control over their supply chains. However, the adoption of new technologies and processes also introduces new risks and challenges, particularly in terms of managing and mitigating these risks.

Furthermore, the growing importance of suppliers in the digital procurement environment makes it imperative for procurement professionals to view suppliers as partners rather than mere vendors. Building mutual trust and collaboration with suppliers is essential to achieve strategic goals and manage risks effectively [3,8].

Additionally, the rapid development of procurement technologies, tools, and software presents both opportunities and challenges for procurement professionals. While these technologies can help streamline procurement processes and enhance efficiency, their adoption requires careful consideration of potential risks and the need for proper training and support for all involved parties [3,9,10].

As organizations embrace digital procurement transformation, there emerges a unique opportunity to advance sustainability goals. The integration of digital technologies allows for enhanced visibility and traceability across the supply chain, facilitating the identification of eco-friendly suppliers and sustainable sourcing practices. Automated processes and data analytics in digital procurement contribute to streamlined operations, reducing resource consumption and minimizing environmental impact. Furthermore, the shift to electronic documentation and communication not only improves efficiency but also aligns with eco-friendly practices by reducing paper usage. In essence, the synergy between digital procurement and sustainability lies in leveraging technological innovations to optimize resource utilization, promote ethical sourcing, and ultimately contribute to environmentally conscious and socially responsible procurement practices.

This research investigates the necessity of risk management in the process of digital procurement transformation. The research was conducted in Serbia using a survey on Google Forms. Respondents were invited through the LinkedIn network without restrictions. As the obtained data are non-quantitative, non-parametric tests—specifically the chi-square, Mann–Whitney U, and Kruskal–Wallis tests—were applied.

This is the first research addressing the necessity of formal risk management in the process of digital procurement transformation in Serbia. The results will aid the academic community in developing a scientific approach to understanding the relationship between risk, the digital transformation of procurement, and sustainability. Procurement cannot fully contribute to sustainability goals without transitioning from traditional to digital practices. Businesses will benefit from this research by receiving concrete guidelines on how to navigate the process of digital procurement transformation. The research was conducted in Serbia, and its tangible contribution to business practice is significant, given that the digital procurement transformation process has just begun. The survey results revealed that only 6.3% of companies can claim their procurement function is fully digitalized, while 20.1% believe they are at some intermediate level of procurement digitalization. An initial stage of procurement digitalization or just a plan for digitalization exists in a substantial 73.6% of companies.

The research results have shown that in the process of digital transformation, attention must be given to the role of procurement professionals, digital technologies, and suppliers as sources of risks. Managing these risks poses a challenge that affects the success of the digital transformation of procurement. Digitalized procurement is a prerequisite for achieving sustainability goals.

## 2. Literature Overview

The role of procurement management has evolved in recent years, particularly with the advent of digital transformation. Companies should have stable, well-established, and supportive top management that has a vision for digital transformation [10]. As organizations continue to embrace digital technologies and processes, the procurement function must adapt and transform to remain relevant and effective in managing risks associated with the procurement process. Digitalization of procurement is a key issue within the supply chain [11]. The digital transformation of procurement is not limited only to large companies [12,13]. Digital transformation brings with it a number of obstacles, and the cost of acceptance can be a limiting factor [14]. The involvement of procurement professionals, digital tools, suppliers, and processes serves as the driving force behind digital transformation [15]. According to Chopra and Sodhi [5], the management of risks is a critical component of procurement management, as it helps identify and mitigate risks that can impact the procurement process, supplier relationships, and, ultimately, organizational performance. Aven [16] emphasizes the importance and need for risk management in the supply chain.

Internal and external factors influence the process of digital procurement transformation and shape it [17]. De la Torre et al. [18], Gottge et al. [19], Hoe [6], Lorentz et al. [17], Rahimi et al. [20], and Moktar et al. [21] state that it is necessary to focus on the development of digitization strategy options for different procurement contexts. Verhoef et al. [22] argue that digital transformation is inevitable, but it should be borne in mind that it carries certain risks by itself. This indicates that digital transformation should not be an end in itself, as deep changes are required that are accompanied by risks.

The importance of risk management in supply chains is recognized both in theory and in practice [8]. Procurement management involves a range of risks, including supply chain disruptions, financial risks, compliance risks, reputational risks, and cyber-security risks. Effective risk management is critical to achieving procurement goals, ensuring compliance with regulations, and maintaining the integrity of the procurement process. According to Flynn et al. [23], the management of risk should be a mandatory part of procurement management, and procurement professionals must be trained to identify, assess, and mitigate risks effectively.

The digital transformation of procurement is a complex process and carries with it a certain level of risk [24]. The implementation of digital solutions is still very demanding and carries with it certain challenges [25]. Organizations need to implement risk mitigation strategies to address specific risks associated with digital procurement transformation. De Boer [4] suggests that these strategies mitigate risks related to cyber-security, data privacy, supply chain disruptions, and vendor management. Chopra and Sodhi [5] emphasize the importance of proactive risk management in supply chains.

Karttunen et al. [26] state that the impact of new technological solutions improves the ability to make decisions about the creation of company strategy and risk management [2]. The digital transformation itself has the effect of reducing maverick buying, but it also creates risks that did not exist until now, and they need to be minimized. Herold et al. [2] point to new opportunities that arise with digital transformation, such as digital analysis of scenarios with the possibility of risk simulation, which should lead to optimization in the use of resources. With increased data transparency, it is possible to improve risk management. Digital procurement solutions can help an organization be more efficient and sustainable [27].

Motaung and Sifolo [25] explore the digital transformation of procurement management and discuss the new skills, knowledge, and capabilities required to effectively manage digital tools and technologies in this field. The authors argue that digital technologies can enhance procurement performance by improving efficiency, effectiveness, and innovation in the procurement process. They also highlight the challenges and opportunities associated with the acceptance of digital procurement technologies.

Costa, and Matias [28], Nicoletti [3], and Ramkumar et al. [29] show the necessity to emphasize the acceptance of innovations and new technologies in the context of procurement. Procurement can be a carrier of digital innovation and thus represents a critical point in the supply chain [30]. Martínez Raya and González-Sánchez [31] say that electronic procurement can lead to the creation of additional risks. Motaung and Sifolo [25] show that the shift from traditional to digital procurement brings many solutions. Digitization enables the reduction of risks in procurement in real time, and procurement is minimally exposed to risks.

Ivanov et al. [32] offer a framework for risk analysis of the new digital supply environment. They try to shift the focus forward and provide a detailed analysis of digital technology in supply chains, as well as the effects of risks arising in the new conditions. New technologies, such as AI, applied in supply chains provide new opportunities for identifying, assessing, and managing risks.

According to Nicoletti [3], there is not much literature in the field of Procurement 4.0, and the available literature is more focused on technical solutions. Procurement 4.0 integrates information and communication technologies and automation to support operational activities. Bienhaus and Haddud [33] explain that the digitalization of procurement requires rethinking the tasks, roles, and responsibilities of all parties in the supply chain and putting in place crossfunctional interdisciplinary systems to speed up transactions and processes in line with the latest technologies. Glas and Kleemann [34] claim that Procurement 4.0, as a digital stage in procurement, with corresponding risk and contract management, can support other company functions to protect the rights of companies within Industry 4.0. According to Nicoletti [3], procurement takes on the new role of team coordinator.

There is a need to develop technologies for a digital supply chain and procurement in a sustainable environment [35]. Digital transformation has great potential for creating and implementing sustainable solutions [36]. Ozkan-Ozen et al. [37] present that Procurement 4.0 contributes to the development of sustainable procurement, and Bag et al. [38], Felsberger and Reiner [39] and Kumar et al. [40] present contributions to ecological, i.e., green, procurement. The use of digital tools has a positive impact on the sustainability of processes in supply chains starting with procurement [41,42]. Changes in procurement caused by digitization can positively affect the optimal use of resources and a sustainable economy [43,44]. Yevu and Yu [45] show that, to facilitate the understanding of eProcurement, it is necessary to understand the factors that drive it.

According to Wang and Pettit [46], digital transformation does not only represent investments in equipment and technology but also in talents, as well as the acquisition of new employee skills. This investment is not a one-off but should be a continuous process to meet the demands of rapidly developing technology. Jumping into digital solutions can be very risky, so in some cases, it is better to first examine the processes, optimize them, and then cover them with digital solutions.

Nicoletti [3] shows that all persons involved in the process must be adequately prepared, informed, and trained for digital procurement solutions. According to Glas and Kleemann [34], the required capabilities of professionals in the procurement function will be changed and may not necessarily require extreme IT skills, as software solutions are expected to become easier to use. Ilhan and Rahim [9] indicate, among other activities in the process of digital transformation, the need for training employees as well as suppliers to successfully respond to new challenges. Adeseun et al. [47] say procurement risks are more easily identified and managed when stakeholders are involved in the process and the consequences are clear. They also emphasized that there is a connection between employees' experience level and the organization itself.

Birou et al. [48] indicate that a gap was identified between the requirements of the industry and school programs. Cagno et al. [49] show that digital technologies in procurement support the improvement of the competence needed to collect, process, and share information, and Zoppelletto et al. [50] claim that with digital transformation, a company

can ensure a fast, clear transfer of knowledge. Tiwari [51] suggests that new research should focus on the skills of employees, needs for training, behavior, work, and barriers to new technology acceptance.

One of the main roles of digitization is increasing visibility and transparency as well as cooperation between partners in supply chains. In his research, Omare [52] provides guidance on how to start digitalization and how to avoid digital mistakes. On the digital path, procurement should take an active role, both within the organization and with external partners—suppliers.

Číž et al. [1] indicate that digitization enables greater visibility and exchange of information between members of the supply chain. The success of digital tools lies in the simplification of processes and clear allocation of responsibilities. This process will mitigate risk and lead to the significant elimination of human error.

Klünder et al. [53] present that digital procurement will strengthen the connection with its suppliers and enable a higher level of collaboration and coordination within procurement activities. In the digital procurement environment, suppliers become partners rather than merely suppliers. According to Kraljic [54], it is necessary to build trust and facilitate effective collaboration between organizations and their suppliers/partners. Bryde et al. [55] mention that building and maintaining strong relationships with suppliers requires effective communication, mutual understanding, and the use of digital procurement tools to facilitate collaboration and data exchange. Nasiri et al. [56] presented that electronic platforms and smart technologies with efficient management lead to better connections with partners and influence the development of procurement. They also pointed out that the growing digital transformation has an impact on the growth of cooperation within the supply channel.

Efficient procurement is one of the key parameters of a company's competitiveness [57], and the use of electronic procurement technologies offers undeniable benefits to supply chains [58]. Based on some research [59–62], the list of technologies used in the digital procurement environment is still open and unexplored. The use of artificial intelligence and machine learning has also found its place in the procurement decision-making process despite some problems in application [63–65]. The use of analytics of large groups of data (Big Data) can serve to identify suppliers globally [66,67] as well as enable better use of supply chains and thus increase the competitiveness of companies [68].

Procurement technologies are developing rapidly, and the most prominent new areas of competence are related to digitization (e.g., eProcurement technology, automation), innovation (e.g., innovative sourcing), and procurement sustainability [69–71]. Digital procurement solutions enable better decisions and improve efficiency [72–74]. Currently, not all firms use procurement software as a single solution for procurement management [72]. The degree of digital maturity has a strong influence on the adoption of digital tools [75]. Leading manufacturers in various industries have already optimized their procurement processes to some extent with new technologies, while small players have not [76].

In conclusion, the digital procurement transformation presents both opportunities and challenges for risk management in procurement. While new technologies and tools offer greater efficiency and effectiveness, they also introduce new risks and uncertainties. To mitigate these risks, procurement professionals must adopt a risk-aware approach that encompasses a range of strategies, including supplier evaluation, contract management, data security, and supply chain visibility. Effective risk management also necessitates collaboration, trust-building, and a skilled workforce. By embracing a holistic and proactive approach to risk management, organizations can successfully navigate the digital procurement landscape and reap the benefits of innovation and digitization.

## 3. Research Methodology

### 3.1. Sample

The sample included 318 respondents, some of whom were members of the Serbian Association of Supply Chain Professionals. A detailed breakdown of the sample's characteristics is provided in Table 1.

**Table 1.** Characteristics of sample.

| Variable | Category | N | % |
|---|---|---|---|
| The main activity of the company | Manufacturing | 157 | 49.37 |
| | Trading | 53 | 16.67 |
| | Services | 62 | 19.50 |
| | Finances | 37 | 11.64 |
| | Other | 9 | 2.83 |
| The size of the company in relation to the number of employees | Less than 50 | 53 | 16.67 |
| | From 50 to 250 | 92 | 28.93 |
| | Over 250 | 173 | 54.40 |
| Annual turnover | Up to EUR 10 million | 142 | 44.65 |
| | From EUR 10 to 100 million | 103 | 32.39 |
| | Over EUR 100 million | 73 | 22.96 |
| Employment sector | Procurement | 136 | 42.77 |
| | Logistics | 36 | 11.32 |
| | Production | 26 | 8.18 |
| | Finance | 63 | 19.81 |
| | Other | 57 | 17.92 |
| Company ownership | Foreign | 155 | 48.74 |
| | Domestic | 163 | 51.26 |

Note: N—number of participants.

The majority of responses came from individuals working in manufacturing companies, accounting for 49.37% of the total. This was followed by a notably smaller proportion of respondents from service-providing companies (19.50%), trading companies (16.67%), and financial institutions (11.64%). Categorizing the research participants based on the size of the companies in which they were employed, 54.40% of the companies have more than 250 employees. Companies employing between 50 and 250 individuals represent 28.93% of the total sample, while those with fewer than 50 employees comprise 16.67%.

Regarding annual turnover, 22.96% of respondents are employed in companies with turnovers exceeding EUR 100 million. This is followed by companies with turnovers ranging from EUR 10 to 100 million, constituting 32.39% of the sample. Lastly, companies with turnovers of up to EUR 10 million account for 44.65% of the responses. The majority of the survey responses originated from procurement professionals, who comprised 42.77% of the total. This was followed by those in finance (19.81%) and logistics (11.32%). Production staff contributed 8.18% of the responses, while the remaining 17.92% came from various other organizational departments. In terms of ownership, 48.7% of the companies involved in the study were foreign-owned, while 51.3% were domestically owned.

*3.2. Procedure*

The study employed online questionnaires created using Google Forms for data collection. Respondent recruitment primarily took place through LinkedIn, reaching beyond the authors' immediate network connections. Additionally, targeted invitations were sent via email to members of the Serbian Association of Supply Chain Professionals.

The data collection phase occurred in the latter half of 2022. All participants were duly informed about the research objectives and gave their consent to participate in the study. Completing the questionnaire required approximately 20 min.

Given the focus on companies operating within Serbia, the questionnaire was administered in Serbian. The choice of geographical focus had a dual rationale: the authors' presence in Serbia and the novelty of such research in the Serbian context. This focus is particularly relevant as Serbia hosts various international companies, potentially reflecting global practices in the findings. Moreover, the presence of foreign companies in Serbia, many of which are implementing digital transformation strategies, adds relevance to the study's results in the context of broader international trends.

### 3.3. Questionnaire

The questionnaire's structure was developed based on an extensive review of relevant literature, identifying four key domains central to digital procurement transformation: procurement processes, the organizational structure of procurement, procurement professionals, and digital technology (including platforms), and suppliers or partners [3]. This framework aimed to explore the interconnections between the sophistication of procurement organizations and various influencing factors.

The questionnaire delved into aspects related to procurement professionals, emphasizing their role in risk management and the emerging requirement for new skill sets in the digital era. The adoption of new technologies and innovation was also a focal point, recognized as a significant parameter in the transformation process. Furthermore, the questionnaire encompassed attitudes related to suppliers, acknowledging their crucial role in the overall efficacy of procurement operations. This comprehensive approach was designed to provide a holistic understanding of the factors driving digital transformation in procurement.

The survey included the following statements, seeking respondents' opinions:

1. Rate the procurement function in your company [25,27,57,58].
2. The role of professional procurement management must include a component of risk management in procurement [5,8,10,15,16,23].
3. Procurement professionals are essential resources in risk management in the procurement business [5,8,10,15,23].
4. All individuals involved in the process must be adequately prepared, informed, and trained to work in a digital environment [3,15,23,25,46,51].
5. Procurement technologies, including tools and software, are developing rapidly [11,13,25,59–62,69–71].
6. There is a need to focus on the acceptance of innovations and new technologies in the context of the purchasing business [3,14,25,28,29].
7. The significance of suppliers in the digital environment is growing, carrying a higher level of risk, leading to them being considered partners rather than merely suppliers [9,26,31,53].
8. Suppliers become partners in the digital procurement environment, emphasizing the importance of building mutual trust [15,30,53,55].
9. Digital tools facilitate the evaluation of suppliers and their qualifications, a significant aspect of procurement management [26,27,32,72–74].
10. Evaluate the state of the purchasing function in your company.

The questionnaire employed a five-point Likert-type scale for all responses. For the initial question, the scale ranged from 1, signifying 'very bad', to 5, indicating 'very good'. Questions two through nine had a response scale ranging from 1, representing 'I don't agree at all', to 5, denoting 'I totally agree'. The tenth question, designed to assess the level of digital transformation, had a response scale ranging from 1, representing 'In the administrative phase—there is no plan for digital transformation', to 5, indicating 'At a high level—fully digitalized'.

### 3.4. Data Analysis

The initial dataset consisted of 321 responses, of which 3 responses were incomplete. Incomplete responses were removed from the dataset before conducting statistical analysis.

Consistent with the hypothesis of this study and the nature of the available data, several statistical analyses were conducted. Pearson's chi-square ($\chi^2$) test was used to examine the association between categorical variables. Regression analysis was applied to assess the relationship between one variable (the criterion variable) and several predictor variables to determine which of the predictors is most strongly associated with the criterion variable. The Kruskal–Wallis test and the Mann–Whitney U test were employed to investigate differences among multiple groups (Kruskal–Wallis test) or between two groups (Mann–Whitney U test) when the dependent variable is categorical. As measures of effect size, the multiple correlation coefficient R and Cramer's V were reported. All analyses were conducted using the SPSS statistical software package v25 (IBM Corp., Armonk, NY, USA, 2017).

## 4. Results

### 4.1. Relationship between the Assessment of the Procurement Function in a Company and Risk Management in Procurement

To examine the relationship between the first and second questions, we conducted Pearson's $\chi^2$ test. The results showed a statistically significant correlation between the assessment of the procurement function (question 1—Q1) and the belief that effective procurement management should include a risk management component (Q2; $\chi^2$ (12) = 109.15, $p < 0.001$, V = 0.338). Predominantly, respondents who partially agreed with this view also rated their procurement function positively. This indicates that in organizations where procurement is viewed favorably, there is an acknowledged need for incorporating risk management.

We used the same approach to explore the link between the first and third questions. The findings revealed a statistically significant correlation between the evaluation of the procurement function (Q1) and the belief in the critical role of procurement professionals in risk management (Q3; $\chi^2$ (12) = 59.22, $p < 0.001$, V = 0.249). Most respondents who strongly agreed with this idea also positively appraised their procurement function. This implies that in organizations with established procurement, the importance of procurement professionals in risk management is more recognized. The crosstabulation table detailing the associations between Q1 and Q2 (upper section) and Q1 and Q3 (lower section) is provided in Appendix A Table A1.

Finally, we conducted a regression analysis to determine which aspect of risk management in procurement (Q2 or Q3) has a stronger association with the evaluation of the procurement function. The regression model was significant (F (2, 315) = 6.78, $p < 0.001$, R = 0.203), indicating that Q2 ($\beta$ = 0.237, $p < 0.05$) has a more pronounced relationship with Q1 compared to Q3 ($\beta$ = −0.041, $p > 0.05$).

### 4.2. Relationship between the Belief That All Persons Involved in the Process Must Be Adequately Prepared, Informed, and Trained to Work in a Digital Environment with Risk Management in Procurement

To investigate the relationship between the fourth and second questions, we applied Pearson's $\chi^2$ test. The analysis revealed a statistically significant correlation between the belief that all participants in the process should be well-prepared, informed, and trained for a digital environment (Q4) and the belief in the necessity of risk management within effective procurement management (Q2; $\chi^2$ (12) = 213.87, $p < 0.001$, V = 0.475). A significant majority of respondents strongly agreed with both viewpoints, highlighting the importance of equipping procurement professionals with digital skills.

We utilized the same method to assess the connection between the fourth and third questions. Our results indicated a statistically significant correlation between the belief in the need for comprehensive training in a digital environment (Q4) and the belief in the pivotal role of procurement professionals in risk management (Q3; $\chi^2$ (12) = 218.22, $p < 0.001$, V = 0.487). Most respondents strongly agreed with both statements. The crosstabulation table that illustrates the relationships between Q4 and Q2 (top section) and Q4 and Q3 (bottom section) is available in Appendix A Table A2.

Finally, a regression analysis was performed to ascertain which aspect of risk management in procurement (Q2 or Q3) is more strongly associated with the belief in the necessity of thorough preparation for a digital environment. The regression model proved to be significant ($F_{(2, 315)} = 81.18$, $p < 0.001$, R = 0.583), showing that Q3 ($\beta = 0.338$, $p < 0.001$) has a marginally stronger association with Q4 than Q2 ($\beta = 0.268$, $p < 0.01$).

### 4.3. Relationship between the Belief That Procurement Technologies Are Developing Rapidly and Risk Management in Procurement

Utilizing the same methodology, we assessed the relationship between Q5 and both Q2 and Q3. We found a statistically significant association between the belief in the rapid development of procurement technologies (Q5) and the necessity of integrating risk management in effective procurement management (Q2; $\chi^2$ (6) = 88.75, $p < 0.001$, V = 0.374). Most respondents somewhat agreed with the former and either somewhat or completely with the latter. Additionally, a statistically significant association was observed between Q5 and the belief in the crucial role of procurement professionals in risk management (Q3; $\chi^2$ (6) = 63.21, $p < 0.001$, V = 0.315). The majority of participants somewhat agreed with the former and completely with the latter. The crosstabulation table displaying the relationships between Q5 and Q2 (upper section) and Q5 and Q3 (lower section) can be found in Appendix A Table A3.

Finally, we conducted a regression analysis to identify which aspect of risk management in procurement (Q2 or Q3) is more closely associated with the belief that procurement technologies are evolving rapidly (Q5). The regression model was significant ($F_{(2, 315)} = 20.84$, $p < 0.001$, R = 0.342), indicating that Q3 ($\beta = 0.313$, $p < 0.01$) has a more substantial association with Q5 compared to Q2 ($\beta = 0.034$, $p > 0.01$).

### 4.4. Relationship between the Belief That It Is Necessary to Focus on the Acceptance of Innovations and New Technologies in the Context of Purchasing Business and Risk Management in Procurement

We observed a statistically significant association between the belief in the necessity of focusing on the acceptance of innovations and new technologies in purchasing (Q6) and the need to integrate risk management in effective procurement management (Q2; $\chi^2$ (9) = 319.84, $p < 0.001$, V = 0.579). Most respondents strongly agreed with both statements. Similarly, there was a significant correlation between Q6 and the belief in the essential role of procurement professionals in risk management (Q3; $\chi^2$ (9) = 418.46, $p < 0.001$, V = 0.662). A majority of participants strongly agreed with both viewpoints. The crosstabulation table illustrating the relationships between Q6 and Q2 (top section) and Q6 and Q3 (bottom section) is available in Appendix A Table A4.

In our final analysis, we conducted a regression analysis to determine which aspect of risk management in procurement (Q2 or Q3) is more strongly linked to the belief in the importance of embracing innovations and new technologies in the purchasing sector (Q6). The regression model was significant ($F_{(2, 315)} = 157.98$, $p < 0.001$, R = 0.708), showing that Q3 ($\beta = 0.440$, $p < 0.001$) has a more pronounced association with Q6 compared to Q2 ($\beta = 0.294$, $p < 0.001$).

### 4.5. Relationship between the Belief That Suppliers Are More Partners Than Suppliers with Trust and Evaluation of Suppliers

To examine the relationship between the perception of suppliers as partners rather than mere suppliers (Q7) and attitudes towards these suppliers in terms of trust (Q8) and evaluation (Q9), we employed $\chi^2$ and regression analyses. The analysis showed a statistically significant relationship between responses to Q7 and Q8 ($\chi^2$ (16) = 502.80, $p < 0.001$, V = 0.629), as well as between responses to Q7 and Q9 ($\chi^2$ (12) = 179.49, $p < 0.001$, V = 0.434). In both instances, most respondents strongly agreed with both questions. The crosstabulation table demonstrating the relationships between Q7 and Q8 (upper section) and Q7 and Q9 (lower section) can be found in Appendix A Table A5.

The regression analysis results indicated a significant model (F (2, 315) = 120.76, $p < 0.001$, R = 0.656), revealing that Q8 (β = 0.534, $p < 0.001$) is more strongly associated with Q7 compared to Q9 (β = 0.193, $p < 0.001$).

*4.6. Differences in the Belief That Effective Procurement Management Should Include a Risk Management Component in the Context of an Assessment of the Procurement Function in a Company*

To evaluate the overall differences in the belief that effective procurement management should incorporate a risk management component, as related to the assessment of the procurement function within a company, we utilized the Kruskal–Wallis test. The results showed persistent overall differences (KW (4) = 39.28, $p < 0.001$). For examining the differences between specific pairs of groups in the assessment of the procurement function, within the context of the aforementioned belief, a series of Mann–Whitney U tests were conducted. These tests revealed no significant differences between the high-level group and the group without a plan for digital transformation (Z = −1.24, $p = 0.216$), between the intermediate-level group and the initial-level group (Z = −1.05, $p = 0.296$), and between the intermediate-level group and the group in which digital transformation is planed (Z = −1.33, $p = 0.183$). However, all other group pairs had significant differences in the context of the belief in the necessity of risk management in procurement, consistent with the group mean rank presented in Figure 1.

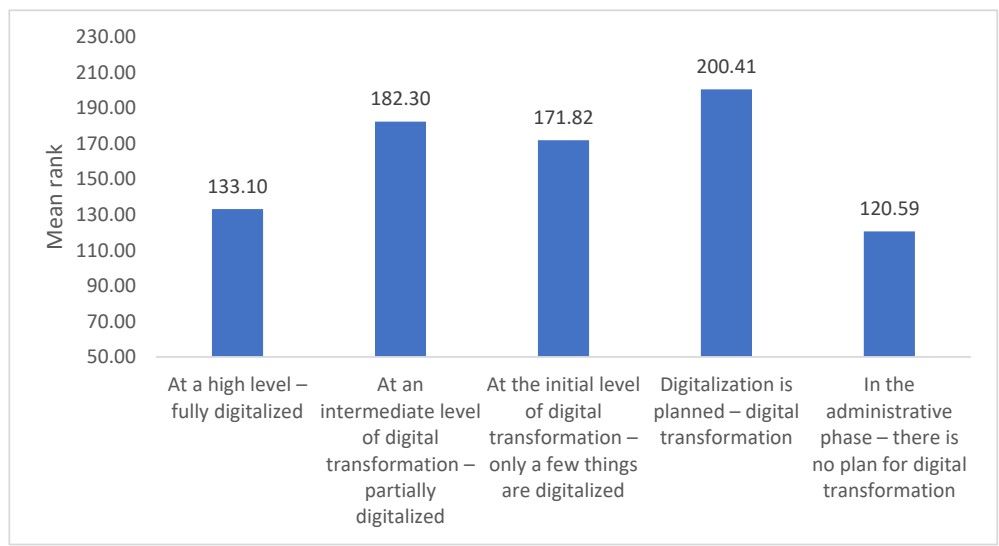

**Figure 1.** Variations in the belief that effective procurement management should include a risk management component, in relation to the assessment of the procurement function within a company.

*4.7. Differences in the Belief That Procurement Professionals Are an Important Resource in Risk Management in the Procurement Business, in the Context of an Assessment of the Procurement Function in a Company*

To assess the overall differences in the belief that procurement professionals are a key resource in risk management within the procurement sector, in relation to the evaluation of the company's procurement function, we employed the Kruskal–Wallis test. The analysis confirmed persistent overall differences (KW (4) = 43.18, $p < 0.001$). To investigate the differences between specific paired groups regarding the procurement function assessment within the context of this belief, we conducted a sequence of Mann–Whitney U tests. These tests showed no significant differences between the high- and intermediate-level groups (Z = −1.96, $p = 0.050$), the high-level group and the group without a digital transformation plan (Z = −0.06, $p = 0.951$), and between the group planning digitalization and both the intermediate-level (Z = −1.81, $p = 0.070$) and initial-level groups (Z = −0.48, $p = 0.628$). Nonetheless, significant differences were found among all other paired groups concerning

the belief in the importance of procurement professionals in risk management, in alignment with the mean group rank depicted in Figure 2.

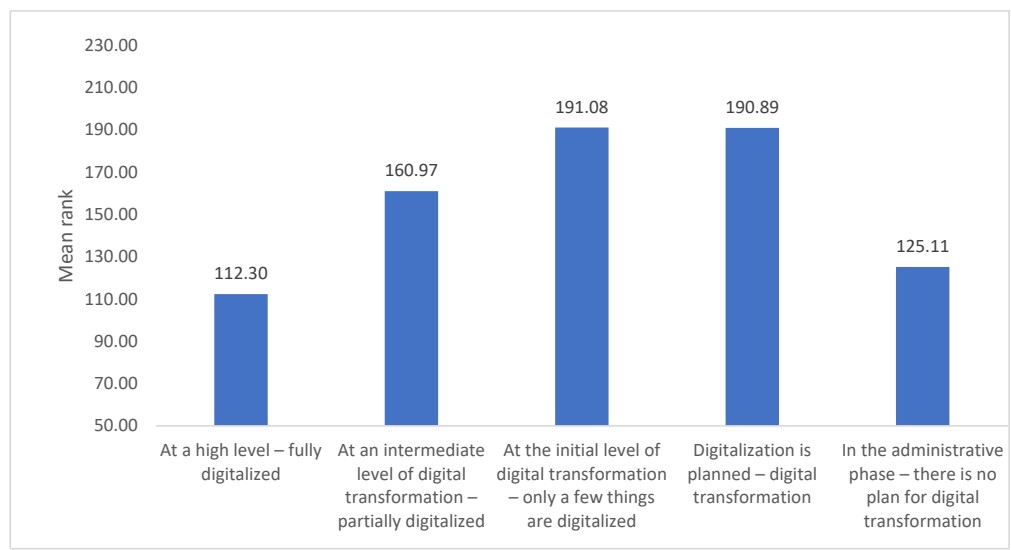

**Figure 2.** Group differences regarding the belief in the importance of procurement professionals in risk management, in relation to the assessment of the procurement function within a company.

## 5. Discussion

The digital transformation of procurement is an ongoing process that cannot be halted [77]. Some companies have already digitized their procurement operations, while others are in the preparation phase or actively engaged in the process. This transformation is perpetual, evolving as digital tools improve, and procurement professionals acquire new knowledge and experience in their roles. Importantly, the digital transformation of procurement is not confined within the company; it involves active participation from suppliers in the procurement interaction [77].

Any change within a company introduces a certain level of risk, and the digital transformation of procurement is no exception. Managing these risks throughout the procurement transformation is essential. Incorrectly handled, this process can jeopardize both the procurement business and the overall company operations [78].

The sample structure in this study offers a nuanced understanding of the interest in the digital transformation of procurement processes across various business sectors and company sizes. The significant representation of manufacturing firms, where procurement complexities are typically more pronounced, highlights a keen interest in digital transformation within this sector. Moreover, the inclusion of responses from diverse industries ensures a comprehensive portrayal of different industry practices and their approach to digital procurement. A notable observation is the predominance of participants from larger enterprises, suggesting a greater inclination towards digital procurement in these organizations. However, this should not overshadow the potential interest in digital strategies among smaller companies, particularly those with lower turnovers, where the efficiency of procurement processes is crucial.

Crucially, the balance in representation between foreign-owned and domestically owned companies in the study underscores its relevance and applicability to both international and local business contexts. This diverse sample composition ensures that the insights derived are reflective of a broad spectrum of business practices, making the study's conclusions robust and globally pertinent.

As such, this research focused on key aspects: procurement professionals, procurement digital technologies, and suppliers, all identified as critical sources of risk. To validate these perspectives, opinions on the quality of the procurement function were sought.

The correlation between confident procurement practices and opinions on the digital transformation of procurement is indicative of a positive direction.

The research was conducted in Serbia, where the formalization of risk management practices is not universal across companies. Notably, risk management has been primarily implemented in companies associated with larger global systems. This broader context ensures that the views expressed in the research are not limited to a specific segment but encompass global best practices. Consequently, the conclusions drawn can be applicable beyond the scope of the research segment.

The primary aim was to establish a connection between risk management, procurement management, and the quality of the procurement function. Given the variance in the adoption of formal risk management practices, confirming this connection became crucial. The findings confirm that successful procurement organizations incorporate risk management into their practices. Subsequently, the research sought to affirm the link between procurement professionals and successful procurement organizations. Procurement professionals emerged as pivotal resources in the digital procurement transformation process, simultaneously posing a potential source of risk. Effective risk mitigation involves thorough preparation, encompassing the development of new skills, access to accurate information, and regular training [7]. Just as the digitization of procurement brings changes, procurement professionals stand as key resources in navigating and shaping these transformations.

When observing the evolution of purchasing technology, rapid growth is evident, accompanied by two types of risk: one stemming from the incorrect selection of tools and the other from their improper utilization [79]. Consequently, a direct link is established between procurement professionals and decision-making regarding tool selection. Remaining current with innovations is essential; however, a willingness to embrace these innovations is equally crucial. Every innovation introduces a certain level of risk, and the ability to accept and control these risks hinges on the preparedness and competence of procurement professionals. Organizations equipped with an established system of risk management in the procurement business are more adept at embracing innovations [80], thereby facilitating the digital procurement transformation process.

Procurement professionals must prioritize innovation and new technologies in the procurement environment [81]. New digital solutions offer avenues to alleviate daily pressures, such as reducing costs, shortening delivery times, and optimizing inventory levels. As the volume of work and environmental challenges expand, procurement organizations must be armed with the appropriate tools to overcome these hurdles.

For procurement professionals to actively engage in risk management, adequate training is imperative [82]. It is confirmed that procurement professionals, serving as a key resource in the digital transformation of procurement, must be prepared to operate in a digital environment. Expecting those accustomed to traditional procurement settings to seamlessly transition into a digital environment without proper training is unrealistic. While procurement processes may echo traditional ones, key differences necessitate identification and preparation.

The digital procurement transformation also encompasses suppliers as a pivotal factor [81]. The growing significance of suppliers in the digital environment elevates the associated risks. Digital procurement tools play a dual role with suppliers: establishing the supply channel and integrating suppliers into the procurement environment, granting them a more substantial role akin to partners. Simultaneously, in the evaluation and qualification of suppliers, digital tools assume a novel role, streamlining the process and enabling better decision-making in supplier selection [83]. Consequently, this has been proven to reduce risk and ensure procurement sustainability.

The research further indicates a heightened inclination toward the need for risk management in procurement and the belief that procurement professionals play a crucial role in procurement risk management, particularly in organizations where the digital procurement transformation is in its early stages. This inclination is rooted in justified concerns about changes, particularly in organizations at the initial phases of digital procurement transfor-

mation or those in the planning stages. Recognizing risk management as a key success factor, organizations understand the imperative of its inclusion in the digital transformation of procurement [84].

Ultimately, the success of the digital transformation of procurement hinges on both digital tools and procurement professionals, as well as suppliers. Each of these factors introduces its own set of risks that can jeopardize the success of the digital procurement transformation and threaten business sustainability. Therefore, a serious and comprehensive approach to risk management during the digitization of procurement is essential.

## 6. Conclusions

A limitation of this research is its focus solely on Serbia. Future research endeavors could extend to neighboring countries. However, it is important to note that the digital transformation of procurement in Serbia initially commenced in companies affiliated with international corporations, headquartered in developed countries. Consequently, foreign practices were applied. An advantage is the presence of companies from numerous developed countries in Serbia, ensuring that the research results offer insights beyond a specific regional practice. The outcomes of this research can benefit medium and small domestically owned enterprises in their digital procurement transformation journey, expediting the process and enhancing the competitiveness of the domestic economy.

The rapid development of digital tools has no influence on the results of this research, as the focus was not on specific tools but on the process itself. The approach to digital procurement transformation must incorporate a risk awareness component, irrespective of the type of digital tool used.

Based on the results, we can conclude that in companies where the procurement organization is well-rated, the role of procurement management must include risk management, and procurement professionals are a vital resource for risk management. This also applies to the belief that procurement technologies are rapidly evolving, emphasizing the need to focus on innovation acceptance.

This paper underscores the significance of a risk-aware approach to digital procurement transformation. The research affirms that a risk-aware approach is crucial in mitigating potential risks associated with new technologies and tools. It emphasizes the importance of training and education for procurement professionals. The findings indicate that all individuals involved in the procurement process must be adequately prepared, informed, and trained to operate in a digital environment. Additionally, the study reveals the growing importance of suppliers/partners in the digital environment, emphasizing the need to establish mutual trust and collaboration.

The research underscores the importance of effective supplier evaluation and selection processes, as well as trust-building mechanisms such as transparent communication, mutual benefits, and shared values. The conclusion drawn is that organizations need to adopt a holistic and proactive approach to risk management, encompassing various strategies such as supplier evaluation, contract management, data security, and supply chain visibility. While potential benefits of digital procurement transformation, including greater efficiency and effectiveness, exist, it is crucial to balance these benefits with the potential risks associated with the digitization of procurement. The research validates the importance of integrating risk management into the context of digital procurement transformation. Through a risk-aware approach and the implementation of diverse strategies, organizations can successfully navigate the digital procurement landscape and capitalize on the benefits of innovation and digitization.

For future researchers, this paper can serve as a foundation for finding a unique approach to the digital transformation of procurement. Currently, there is no singular and proven approach, which likely contributes to failures in the digital transformation process.

Further research suggestions:

1.  Investigate the effectiveness of different risk management strategies in the context of digital procurement transformation. For instance, explore the most effective methods

for evaluating and selecting suppliers/partners in a digital environment, and examine how organizations can establish trust and collaboration with their partners.

2.  Explore the role of emerging technologies such as artificial intelligence, blockchain, and the Internet of Things (IoT) in digital procurement transformation. Assess the potential benefits and risks associated with these technologies, and examine effective integration strategies into procurement processes.

3.  Conduct a comparative study of risk management practices across different industries and sectors. Analyze variations in risk management practices and identify lessons that organizations can learn from other sectors.

4.  Investigate the impact of digital procurement transformation on supply chain sustainability and social responsibility. Explore how organizations can leverage digital technologies to promote sustainability and responsible sourcing practices.

5.  Conduct a longitudinal study to assess the long-term impact of digital procurement transformation on organizational performance and competitiveness. Examine how the benefits and risks evolve over time and identify key success factors for sustained performance improvement.

**Author Contributions:** Conceptualization, Ž.D. and V.V.; methodology, Ž.D., V.V., V.P. and I.E.; software, Ž.D. and V.V.; validation, Ž.D., I.E. and S.V.; investigation, Ž.D. and D.D.; data curation, Ž.D.; writing—original draft preparation, Ž.D., V.V. and S.V.; writing—review and editing, Ž.D., V.V., I.E., D.D., V.P. and S.V.; visualization, Ž.D., D.D. and I.E.; supervision, V.V., V.P. and S.V. All authors have read and agreed to the published version of the manuscript.

**Funding:** This research received no external funding.

**Institutional Review Board Statement:** Not applicable.

**Informed Consent Statement:** Not applicable.

**Data Availability Statement:** The data presented in this study are available on request from the corresponding author.

**Acknowledgments:** The authors thank the anonymous reviewers for their valuable comments and suggestions.

**Conflicts of Interest:** The authors declare no conflicts of interest.

## Appendix A

**Table A1.** Combination 1 with 2 and 1 with 3.

| Questions | | 1. Rate the procurement function in your company on a scale of 1 to 5, with 1 being the worst and 5 being the best | | | | | Total |
|---|---|---|---|---|---|---|---|
| | Answer | Worst grade | Bad grade | Middle grade | Good grade | Best grade | |
| 2. The role of professional procurement management must include risk management in procurement management. | I partially disagree | 0 | 9 | 2 | 0 | 0 | 11 |
| | I do not know | 18 | 0 | 16 | 11 | 0 | 45 |
| | I partially agree | 0 | 35 | 40 | 52 | 0 | 127 |
| | I totally agree | 10 | 36 | 33 | 47 | 9 | 135 |
| Total | | 28 | 80 | 91 | 110 | 9 | 318 |
| 3. Procurement professionals are an important resource in risk management in the procurement business. | I partially disagree | 0 | 9 | 0 | 0 | 0 | 9 |
| | I do not know | 9 | 0 | 18 | 16 | 0 | 43 |
| | I partially agree | 9 | 27 | 29 | 28 | 0 | 93 |
| | I totally agree | 10 | 44 | 44 | 66 | 9 | 173 |
| Total | | 28 | 80 | 91 | 110 | 9 | 318 |

**Table A2.** Combination 4 with 2 and 4 with 3.

| Questions | | 4. All persons involved in the process must be adequately prepared, informed, and trained to work in a digital environment | | | | | Total |
|---|---|---|---|---|---|---|---|
| | Answer | I don't agree at all | I partially disagree | I do not know | I partially agree | I totally agree | |
| 2. The role of professional procurement management must include risk management in procurement management. | I partially disagree | 0 | 1 | 9 | 0 | 1 | 11 |
| | I do not know | 9 | 8 | 9 | 1 | 18 | 45 |
| | I partially agree | 0 | 0 | 18 | 52 | 57 | 127 |
| | I totally agree | 0 | 0 | 0 | 27 | 108 | 135 |
| Total | | 9 | 9 | 36 | 80 | 184 | 318 |
| 3. Procurement professionals are an important resource in risk management in the procurement business. | I partially disagree | 0 | 0 | 9 | 0 | 0 | 9 |
| | I do not know | 9 | 9 | 8 | 8 | 9 | 43 |
| | I partially agree | 0 | 0 | 10 | 35 | 48 | 93 |
| | I totally agree | 0 | 0 | 9 | 37 | 127 | 173 |
| Total | | 9 | 9 | 36 | 80 | 184 | 318 |

**Table A3.** Combination 5 with 2 and 5 with 3.

| Questions | | 5. Procurement technologies—tools and software are developing rapidly | | | Total |
|---|---|---|---|---|---|
| | Answer | I do not know | I partially agree | I totally agree | |
| 2. The role of professional procurement management must include risk management in procurement management. | I partially disagree | 10 | 1 | 0 | 11 |
| | I do not know | 43 | 0 | 2 | 45 |
| | I partially agree | 29 | 54 | 44 | 127 |
| | I totally agree | 45 | 54 | 36 | 135 |
| Total | | 127 | 109 | 82 | 318 |
| 3. Procurement professionals are an important resource in risk management in the procurement business. | I partially disagree | 9 | 0 | 0 | 9 |
| | I do not know | 35 | 0 | 8 | 43 |
| | I partially agree | 38 | 36 | 19 | 93 |
| | I totally agree | 45 | 73 | 55 | 173 |
| Total | | 127 | 109 | 82 | 318 |

**Table A4.** Combination 6 with 2 and 6 with 3.

| Questions | | 6. It is necessary to focus on the acceptance of innovations and new technologies in the context of purchasing business | | | | Total |
|---|---|---|---|---|---|---|
| | Answer | I partially disagree | I do not know | I partially agree | I totally agree | |
| 2. The role of professional procurement management must include risk management in procurement management. | I partially disagree | 9 | 2 | 0 | 0 | 11 |
| | I do not know | 17 | 27 | 1 | 0 | 45 |
| | I partially agree | 0 | 9 | 80 | 38 | 127 |
| | I totally agree | 9 | 8 | 12 | 106 | 135 |
| Total | | 35 | 46 | 93 | 144 | 318 |
| 3. Procurement professionals are an important resource in risk management in the procurement business. | I partially disagree | 9 | 0 | 0 | 0 | 9 |
| | I do not know | 8 | 35 | 0 | 0 | 43 |
| | I partially agree | 9 | 1 | 72 | 11 | 93 |
| | I totally agree | 9 | 10 | 21 | 133 | 173 |
| Total | | 35 | 46 | 93 | 144 | 318 |

**Table A5.** Combination 7 with 8 and 7 with 9.

| Questions | Answer | 7. The importance of suppliers in the digital environment has growing importance and carries with it a higher level of risk, so we can call them partners rather than suppliers | | | | | Total |
|---|---|---|---|---|---|---|---|
| | | I don't agree at all | I partially disagree | I do not know | I partially agree | I totally agree | |
| 8. Suppliers become partners in the digital procurement environment and for this reason, it is important to build mutual trust. | I don't agree at all | 9 | 0 | 0 | 0 | 0 | 9 |
| | I partially disagree | 0 | 17 | 0 | 8 | 0 | 25 |
| | I do not know | 0 | 9 | 36 | 28 | 1 | 74 |
| | I partially agree | 0 | 1 | 17 | 17 | 29 | 64 |
| | I totally agree | 0 | 8 | 19 | 27 | 92 | 146 |
| Total | | 9 | 35 | 72 | 80 | 122 | 318 |
| 9. Digital tools facilitate the evaluation of suppliers and their qualifications, which is a significant part of procurement management. | I partially disagree | 0 | 8 | 9 | 0 | 0 | 17 |
| | I do not know | 0 | 18 | 26 | 1 | 10 | 55 |
| | I partially agree | 9 | 0 | 11 | 43 | 21 | 84 |
| | I totally agree | 0 | 9 | 26 | 36 | 91 | 162 |
| Total | | 9 | 35 | 72 | 80 | 122 | 318 |

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
