# Peer review of "A Risk-Aware Approach to Digital Procurement Transformation"

_sustainability, doi:10.3390/su16031283_

Round 1

Reviewer 1 Report

Comments and Suggestions for Authors

The paper should be seriously improved from the Introduction until the Conclusion section.

My additional comments are the following:

The subject of the paper is interesting, but the approach needs improvements as follows:

We found no hypothesis or research question, even in the paper on p. 6, r. 270-271 is mentioned, ‘The chi-square distribution is commonly employed in hypothesis testing to compare 270 observed data with expected data.’ in the Abstract section, it is mentioned, ‘The research questions were formulated to scrutinize the imperative role of risk management in digital procurement, as well as the essential preparation and training required for individuals involved in the process.’

There is a difference between research questions and questions in the questionnaire.

Materials and Methods sections should be improved by adding more details regarding the research data collection, analysis, and interpretation of results.

The authors should briefly describe the methods of data employed and their application and appropriateness for data analysis.

On p. 6, r. 275 is mentioned, ‘The questionnaire included the following questions, seeking respondents' opinions:’, but below, from an editing point of view, are sentences and not questions.

The results were not well-presented to readers to understand the focus of the research study.

Discussing the results could be improved by interpreting them to support theories related to the research topic.

A rewrite is advised to improve clarity. One way to do this is to move the explanations of what is shown in the nine tables and place them before the tables. The explanations need to tell a story about the key findings in each table, rather than just repeating what is shown, and include the researchers’ interpretation. It would also help to reduce the number of Tables.

The authors' names should be mentioned correctly in the article's body. For example, p. 3, r. 101: ‘S. Chopra and M.S. Sodhi [5]’ correct is ‘Chopra and Sodhi [5]’.

Good luck!

Comments on the Quality of English Language

The English language needs improvements.

Author Response

Dear Reviewer,

Thank you for reviewing our manuscript,

Please see the attachment,

Thanks in advance,

Best regards,

Authors

Reviewer 2 Report

Comments and Suggestions for Authors

See attached file

Comments on the Quality of English Language

I suggest reviewing the English and simplify some sentences.

Author Response

(The authors gave the same response as above.)

Reviewer 3 Report

Comments and Suggestions for Authors

Thank you for the opportunity to review the article entitled ‘A Risk-Aware Approach to Digital Procurement Transformation’, which is cognitively interesting and addresses important and current research issues. In my opinion, the paper requires a few improvements - first of all, the methods section, including considering how to formulate the research objective(s) (I believe that posing detailed research questions will improve the considerations), how the authors developed the research tool, the characteristic of research sample and to characterize the following research steps. My comments are indicated below.

1)      The text of the article require proof reading. Some sections of the paper/words are highlighted in red, it is not clear for what reason. Has it been improved after a proofreading?

2)      The abstract could be more concise and more clearly organized. For example, in line 13 the authors refer to what the questions are about, and in lines 19-20 they again refer to the research methodology.

3)      The introduction indicates the purpose of the manuscript, which is:’ The goal of this paper is to investigate the need for including risk management in the process of digital procurement transformation.’  In my opinion it is not convincing as a purpose of a research paper. Perhaps if the authors had pointed to specific research questions/objectives in doing so it would have been clearer to a reader. 

4)      It is not clear how the survey tool was constructed. It is also not clear from the literature review, which also could have been better organized with respect to the themes raised. The authors did not provide characteristics of the research sample, and it is not clear who participated in the study. Could the results have been different if other subjects had been invited to participate in the study? In what way? What are the authors' conclusions in this area? The research steps should be described in more detail.

5)      The methods part should be improved. The Authors indicate (305-306): ‘After collecting the responses, the data analysis process commenced. Initially, a combination of questions 10 and 1 was scrutinized.’ But why 10 and 1 first; next 10 and 2 and then 7 and 1? Have the authors conducted additional tests to prove differences in the strength of particular correlations? Did the authors also analyze the median for the analyses shown in Table 11 and 12?

6)      The discussion section lacks references to previous findings in the literature. It is unclear where the authors indicate their own thoughts and where they refer to the considerations of other researchers.

7)      Theoretical contributions should be more emphasized.

Comments on the Quality of English Language

The paper is linguistically satisfying. I think that after re-reading, the authors will catch the shortcomings. However, it is not clear why some words are highlighted in red.

Author Response

(The authors gave the same response as above.)

Reviewer 4 Report

Comments and Suggestions for Authors

The list of references does not have uniform criteria and does not respect the format of the journal. Particularly in journal references, writing criteria and volume and issue format.

Author Response

(The authors gave the same response as above.)

Round 2

Reviewer 1 Report

Comments and Suggestions for Authors

The paper was improved, but still needs improvement.

We suggest paying attention to methodology, results, and discussion.

We think that the methodology still is the main weakness of the paper.

For example, the tables from Table 1 to Table 7 are not at the scientific level of an article. 

A discussion about the literature should follow every table.

The article contains 85 sources, but these sources are presented without a critical analysis. Moreover, the sources are not logically presented. For example, the source [12] is mentioned on p. 5, r. 242 after the source [69].

Good luck!

Comments on the Quality of English Language

The English language was improved.

Author Response

Dear Reviewer,

Thank you for your suggestions to improve our text. We have incorporated them and made the necessary corrections. A detailed response is provided in the attached file.

Thanks again,

Best Regards, Authors

Reviewer 2 Report

Comments and Suggestions for Authors

Accept in present form

Comments on the Quality of English Language

 Moderate editing of English language required

Author Response

Dear reviewer,

Thank you for your suggestions to improve our text. 

Following the guidance of all reviewers, the major revisions in our updated manuscript include:

  • Language improvements - ensuring even greater clarity and refinement;
  • Research methodology - now presented more clearly, detailed, and concisely;
  • Research results, which have been thoroughly revised to meet the journal's standards. We have modified the presentation of results and added several analyses to better align with the research aims;

Once again, we thank you for your time and insights that have enriched the quality of our paper. We hope that the manuscript is now in a form suitable for publication in this esteemed journal.

Thanks again,

Best Regards,

Authors

Round 3

Reviewer 1 Report

Comments and Suggestions for Authors

Good luck!